# Molecular Modeling and Experimental Evaluation of Non-Chiral Components of Bergamot Essential Oil with Inhibitory Activity against Human Monoamine Oxidases

**DOI:** 10.3390/molecules27082467

**Published:** 2022-04-11

**Authors:** Raffaella Catalano, Francesca Procopio, Daniel Chavarria, Sofia Benfeito, Stefano Alcaro, Fernanda Borges, Francesco Ortuso

**Affiliations:** 1Department of Health Sciences, University “Magna Graecia” of Catanzaro, Campus Universitario “S. Venuta”, Viale Europa, Loc. Germaneto, 88100 Catanzaro, Italy; catalano@unicz.it (R.C.); francesca.procopio001@studenti.unicz.it (F.P.); alcaro@unicz.it (S.A.); 2Net4Science s.r.l., University “Magna Graecia” of Catanzaro, Campus Universitario “S. Venuta”, Viale Europa, Loc. Germaneto, 88100 Catanzaro, Italy; 3CIQUP/Department of Chemistry and Biochemistry, Faculty of Sciences, University of Porto, 4169-007 Porto, Portugal; danielf_chavarria@live.com.pt (D.C.); sofia_benfeito@hotmail.com (S.B.); mfernandamborges@gmail.com (F.B.); 4CRISEA—Centro di Ricerca e Servizi Avanzati per l’Innovazione Rurale, 88055 Belcastro, Italy

**Keywords:** *h*MAOs, Parkinson’s disease, bergamot essential oil, bergamottin, molecular docking, molecular dynamics

## Abstract

Human monoamine oxidases (*h*MAOs) are well-established targets for the treatment of neurological disorders such as depression, Parkinson’s disease and Alzheimer’s disease. Despite the efforts carried out over the years, few selective and reversible MAO inhibitors are on the market. Thus, a continuous search for new compounds is needed. Herein, MAO inhibitors were searched among the non-chiral constituents of Bergamot Essential Oil (BEO) with the aid of computational tools. Accordingly, molecular modeling simulations were carried out on both *h*MAO-A and *h*MAO-B for the selected constituents. The theoretically predicted target recognition was then used to select the most promising compounds. Among the screened compounds, Bergamottin, a furocoumarin, showed selective *h*MAO-B inhibitory activity, fitting its active site well. Molecular dynamics simulations were used to deeply analyze the target recognition and to rationalize the selectivity preference. In agreement with the computational results, experimental studies confirmed both the *h*MAO inhibition properties of Bergamottin and its preference for the isoform B.

## 1. Introduction

Monoamine oxidases (MAO, EC 1.4.3.4) are privileged molecular targets in neuroscience because of their pivotal role in modulating the levels of monoamine neurotransmitters [1].

MAOs are mitochondrial flavoenzymes that catalyze the oxidative deamination of endogenous and exogenous monoamines. MAOs terminate the actions of the amine neurotransmitter serotonin (5-hydroxytryptamine, 5-HT), dopamine (dopamine, DA), noradrenaline (norepinephrine, NE) and adrenaline (epinephrine, EP). Therefore, they play an important role in behavioral and cognitive functions and are related to the development of neurological disorders, including depression, Parkinson’s disease and Alzheimer’s disease [2,3]. 

Two distinct isoforms are encoded in the human genome: MAO-A and MAO-B. Although they share 73% of their sequence identity, human MAO (*h*MAO) isoforms differ in tissue distribution, substrate specificity and inhibitor selectivity. *h*MAO-A is dominant in the placenta, gastrointestinal tract and heart; exhibits higher affinity toward serotonin; and is inhibited by the acetylenic inhibitor Clorgyline. MAO-B can be found in glial cells, platelets and the liver; catalyzes the oxidation of benzylamine and 2-phenylethylamine; and is inhibited by l-Deprenyl. Catecholamines are substrates for both *h*MAOs isoforms. With aging, the MAO-B activity increases in the human brain and correlates, in the *substantia nigra*, with a higher Dopamine (DA) oxidation. The increased levels of oxygen-free radicals derived from the DA metabolism trigger the dopaminergic neuron damage underlying Parkinson’s disease [4]. 

The X-ray structure resolution of *h*MAOs provided insights into the molecular aspects underlying the preference for specific substrates and different inhibitor affinities. Small differences between *h*MAO-A and *h*MAO-B active sites are responsible for the overcited specificity. Both *h*MAO isoforms share the overall fold with most of the residues conserved [5,6]. The active site, which hosts the *h*MAO inhibitors, extends in front of the flavin adenine dinucleotide (FAD) cofactor and is a single hydrophobic cavity of about 400 Å^3^ in *h*MAO-A and a double elongated cavity of about 700 Å^3^ in *h*MAO-B. The non-conservative residues Asn181, Phe208 and Ile335 in *h*MAO-A (Cys172, Ile199 and Tyr326, respectively in *h*MAO-B) are responsible for the differences in the active site shape. The Ile199 in *h*MAO-B mainly contribute to the bipartite nature of the cavity [7,8].

Over the last few years, great efforts have been made to develop potent MAO inhibitors (MAOIs). The selective inhibition of *h*MAO-A is a well-established approach for the mental disorder treatment, whereas selective *h*MAO-B inhibitors are used to treat Parkinson’s disease [9,10]. A continuous search for MAOIs is needed because of their low selectivity and irreversibility.

In this study, a selection of bergamot essential oil (BEO) non-chiral compounds was screened against *h*MAO-A and *h*MAO-B with the aim of identifying novel selective MAOIs. BEO is the product of the mechanical manipulation of the exocarp (flavedo) of the *Citrus bergamia Risso* (“Bergamot”), belonging to the *Rutaceae* family, which grows almost exclusively in Calabria (Italy) [11]. Bergamot essential oil is widely used in perfumes, cosmetics, as well as in the food and confectionery industry. BEO also presents antimicrobial properties [12] and has been investigated for its potential antiproliferative effects [13]. Moreover, several studies described BEO’s effects in chronic psoriasis and in aromatherapy to reduce stress-induced anxiety [14].

The chemical composition of BEO was deeply investigated and characterized. The volatile fraction represents the 93–96% of the total content. Limonene, γ-Terpinene, Linalyl acetate, β-Pinene and Linanalol mainly contribute to the volatile fraction composition. The non-volatile fraction only represents 4–7% of the total content. The most relevant constituents are oxygen heterocyclic compounds, including coumarins and psoralens [15,16].

Among the screened compounds, bergamottin exhibited *h*MAOs inhibition properties both in in silico and in the experimental assays, providing a rational basis for further investigation.

## 2. Results and Discussion

### 2.1. In Silico Monoamine Oxidase Inhibition Studies

The BEO chemical constituents were investigated for the identification of new *h*MAO inhibitors. Already known MAOIs and chiral compounds were excluded from the selection. In fact, chiral molecules introduce uncertainty if they are not available as pure enantiomers: molecular modeling results will be related to the single enantiomer while experimental data will be derived from the mixture. The final selection (Materials and Methods) included a furocoumarin (Bergamottin) and three flavons (Nobiletin, Sinensetin and Tangeritin). Molecular modeling studies were carried out to inspect the activity and selectivity of the selected compounds toward *h*MAO-A and *h*MAO-B. The interaction with the targets was preliminarily estimated by means of molecular docking. The Glide docking score (GScore) of the top ranked pose is reported in Table 1. Among the investigated compounds, only Bergamottin recognized the catalytic site of both *h*MAO isoforms, showing a better GScore against *h*MAO-B (−8.09 Kcal/mol) than against *h*MAO-A (−6.23 Kcal/mol). Nobiletin, Sinensetin and Tangeritin were unable to recognize the known active site of *h*MAOs, resulting in a low binding affinity. Graphical inspection showed they can mildly interact on a different site, away from the active site.

To deeply analyze the Bergamottin recognition of MAOs active sites, the top-ranked docking complexes in *h*MAO-A and in *h*MAO-B were subjected to Molecular dynamics (MD) simulation (Materials and Methods). In both cases, the starting geometry was stable during the simulation time. Bergamottin was accommodated in the *h*MAO-A binding site with the psoralen portion oriented perpendicular to the FAD cofactor and the 3,7-dimethylocta-2,6-dienoxy alkyl chain extended towards the active site entrance. The psoralen ring was found in π-π contact with Tyr407 and Phe352 in 85% and 74% of the MD trajectory frames sampled, respectively. The same interaction, with a frequency equal to 20%, was observed with respect to Tyr444. Interestingly, Bergamottin psoralen moiety established hydrogen bonds (HB) with explicit solvent molecules in 13% of the MD-generated structures. The ligand alkyl chain was involved in hydrophobic contact with Ala111, Ile180, Phe208, Val210, Ile335 and Leu337 in at least 10% of the trajectory frames (Figure 1a). Due to induced fit phenomena allowed by MD, an unfavorable electrostatic repulsion between Asn181 and the Bergamottin side chain sp3 oxygen linker, initially observed in the docking pose, disappeared.

In *h*MAO-B, the psoralen ring of bergamottin highlighted a larger π-π stacking network, both in terms of involved residues and frequency. In fact, such productive interactions were observed with Tyr326 (68%), Phe343 (53%), Tyr398 (88%) and Tyr435 (30%). In opposition to the *h*MAO-A case, HB were highlighted between the furan ring of Bergamottin and Tyr60 (9%) and between the ligand side-chain sp3 oxygen and Cys172 (8%). As previously reported for the *h*MAO-A, and also in *h*MAO-B, hydrophobic contacts among Bergamottin alkyl moiety and Phe168, Leu171, Ile199 and Ile 316 were highlighted in at least the 10% of the MD-sampled structures (Figure 1b).

Overall, the abovementioned interactions may be addressed to rationalize the productive recognition of Bergamottin with respect to MAOs. On the other hand, the replacement of *h*MAO-B Cys172—involved in HB with Bergamottin—by Asn181 in *h*MAO-A, which did not productively interact with the ligand, and the active site solvent exposition, which was wider in *h*MAO-A than in *h*MAO-B, may explain the target isoform selectivity.

### 2.2. In Vitro Monoamine Oxidase Inhibition Studies

Based on the data obtained from in silico studies, Bergamottin was selected to follow through to the experimental evaluation of its *h*MAO-A and *h*MAO-B inhibitory potency (IC_50_) and selectivity (selectivity index, SI). To this end, a spectrophotometric method based on the oxidative deamination of Kynuramine by recombinant *h*MAO-A or *h*MAO-B was used. The reference inhibitors Clorgyline (*h*MAO-A) and Rasagiline (*h*MAO-B) were also included in this study for comparative purposes. The results obtained are reported in Table 2.

Bergamottin inhibited both *h*MAO isoforms, showing a higher preference for isoform B (SI = 31.8). Indeed, Bergamottin presented a *h*MAO-B IC_50_ value within the nanomolar range (291 nM) and a *h*MAO-A IC_50_ value within the micromolar range (9.25 µM). Moreover, both the *h*MAOs IC_50_ values and selectivity indexes of Bergamottin were within the same range of the reference *h*MAO-B inhibitor Rasagiline.

Overall, these results are in line with the data obtained in the molecular modeling studies that showed higher binding affinity of Bergamottin for *h*MAO-B than for *h*MAO-A (Table 1).

## 3. Materials and Methods

### 3.1. Molecular Modeling

The 3D structures of BEO non-chiral and unknown iMAOs compounds were retrieved from PubChem Database (Bergamottin, CID: 5471349; Nobiletin, CID: 72344; Sinensetin, CID: 145659; Tangeretin, CID: 68077) [17] and processed with LigPrep [18] tool to define the protonation states at physiological pH. The crystallographic models of *h*MAO-A and *h*MAO-B with the codes 2Z5X [8] and 6FW0 [19], respectively, were obtained from the Protein Data Bank (PDB) [20]. The original PDB X-Ray structures were suitably optimized with the Protein Preparation Wizard tool [21], using the OPLS-2005 force field [22]. In detail, hydrogen atoms were added, the correct bond orders were assigned, and missing atoms side-chains and loops were built. Moreover, the co-crystallized water molecules were removed, and the FAD connectivity was fixed. A docking grid box, with a volume equal to 64,000 Å^3^, was built using the co-crystallized PDB ligands, Harmine for *h*MAO-A and chlorophenyl–chromone–carboxamide for *h*MAO-B, using Glide software [23]. The regular receptor grid was centered on the FAD N5 atom. The binding affinity was estimated by means of the Glide GScore scoring function. The docking exploration was carried out with the standard precision (SP) Glide search algorithm and the ligands’ conformational flexibility was taken into account.

The best docking pose of Bergamottin into *h*MAO-A and *h*MAO-B was subjected to Molecular Dynamics (MD) simulation using the Desmond program [24,25]. Water solvent effects were mimicked by means of the SPC [26,27] explicit solvation model. In detail, 63,195 and 48,834 water molecules were added to the *h*MAO-A and *h*MAO-B Bergamottin top-ranked docking complexes, respectively. The overall net charge was neutralised by including 2 Cl^−^ and 3 Na^+^ counterions to the explicitly solvated *h*MAO-A and *h*MAO-B models, respectively. The starting structures built thus were submitted to the Desmond default relaxation protocol. The production MD runs were carried out up to 10 ns, with an integration time step equal to 2 fs, at 300 °K. MD trajectories were sampled at regular intervals equal to 100 ps, collecting 100 conformers for each system. In order to prevent unrealistic distortion of atoms position, the SHAKE algorithm [28,29] was applied to all hydrogens. All sampled structures were considered for investigating, by means of the namesake Desmond tool, the ligand target interaction analysis. These descriptors were computed, on the basis of the complex geometry, on each sampled structure and their frequency was reported as a percentage.

### 3.2. In Vitro Monoamine Oxidase Inhibition Studies

The inhibitory activity of Bergamottin on *h*MAO-A and *h*MAO-B was studied using an experimental protocol described elsewhere. [30,31] The *h*MAO inhibitory activity was assessed in microsomal MAO isoforms prepared from insect cells (BTI-TN-5B1-4) infected with recombinant baculovirus containing cDNA inserts for *h*MAO-A or *h*MAO-B, and by measuring the enzymatic conversion rates of Kynuramine into 4-hydroxyquinoline. The appropriate amounts of *h*MAO-A and *h*MAO-B were adjusted to obtain, in our experimental conditions, the same maximum velocity (Vmax = 50 pmol/min) for both isoforms (*h*MAO-A: 3 ng/µL; hMAO-B: 12 ng/µL). All assays were performed in sodium phosphate buffer solution 50 mM pH 7.4.

Bergamottin or the reference MAO inhibitors were preincubated at 37 °C for 10 min in the presence of Kynuramine (*K*_m_ *h*MAO-A = 20 µM; *K*_m_ hMAO-B = 20 µM; final concentration: 2 × *K*_m_) in 96-well microplates (BRANDplates, pureGradeTM, BRAND GMBH, Wertheim, Germany). Then, the reaction was started with the addition of *h*MAO-A or *h*MAO-B. Initial velocities were determined spectrophotometrically in a microplate reader (BioTek Synergy HT from BioTek Instruments, Winooski, VT, USA) at 37 °C by measuring the formation of 4-hydroxyquinoline at 316 nm, over a period of at least 30 min (interval of 1 min). Data were analyzed using GraphPad PRISM version 6 for Windows (GraphPad Software^®^, San Diego, CA, USA). The initial velocities, obtained from the linear phase of product formation, were normalized and plotted against the respective inhibitor concentration. IC_50_ values were obtained from dose–response curves and were expressed as mean ± standard deviation. IC_50_ values were determined from at least three independent experiments, each performed in triplicate. Microsomal MAO isoforms, Kynuramine, Rasagiline and Clorgyline were purchased from Sigma-Aldrich (St. Louis, MO, USA). Bergamottin, Nobiletin, Sinensetin, and Tangeritin were purchased from Titolchimica s.p.a. (Pontecchio Polesine, Italy).

## 4. Conclusions

In this work, in silico studies and biological assays were performed to explore the potential inhibition of BEO selected compounds towards *h*MAOs. In particular, Bergamottin was demonstrated to be a promising *h*MAO inhibitor with higher selectivity for *h*MAO-B. Indeed, the presence of stacking, hydrogen bonds and hydrophobic interactions between the compound and the target, and the interesting binding mode assumed within the active site of the enzyme, may provide a rational explanation for its selectivity. The theoretical results were in accordance with the experimental data obtained in *h*MAOs inhibition assays. Finally, this study discovered a scientifically unknown activity of Bergamottin, suggesting a tentative new application for BEO. In fact, taking into account that MAO enzymatic activity increases radical oxygen species levels at the central nervous system, the administration of BEO—or Bergamot, as a nutraceutical—could limit neuronal oxidative stress with beneficial effects in the prevention of high-social-impact neurodegenerative disorders. Moreover, considering our results, Bergamottin could be considered as a leading compound for the development of novel selective *h*MAOs inhibitors.

## Figures and Tables

**Figure 1 molecules-27-02467-f001:**
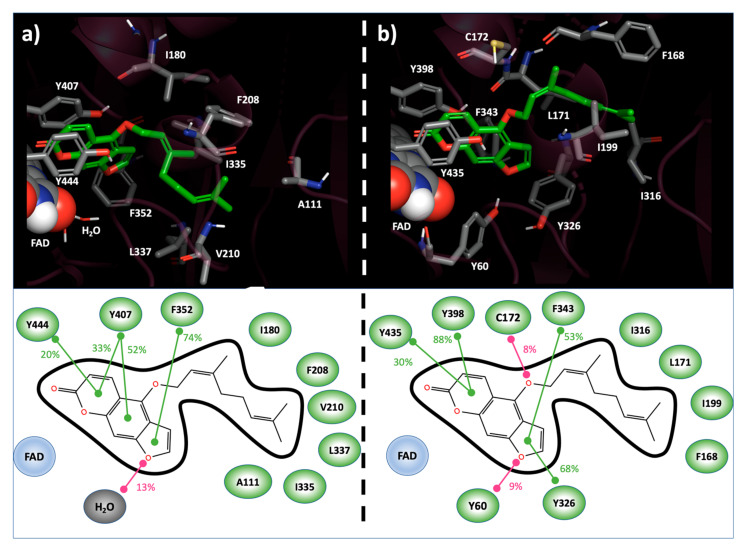
Bergamottin binding modes into the *h*MAO-A (**a**) and *h*MAO-B (**b**) active site resulting from MD simulation. In 3D representation (**top**), interacting residues and Bergamottin are reported in CPK and green-carbon-colored sticks, respectively. FAD cofactor is represented as spheres. In 2D scheme (**bottom**), ligand–target interactions, monitored during the MD simulation, are reported. Stacking contribution are depicted as green lines, hydrogen bonds as magenta lines. Frequencies of each interaction are indicated as percentages.

**Table 1 molecules-27-02467-t001:** Docking GScore value of selected BEO non-chiral compounds screened against *h*MAO-A and *h*MAO-B.

ID Name	2D Structure	GScore (Kcal/mol)
*h*MAO-A	*h*MAO-B
Bergamottin	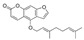	−6.23	−8.09
Nobiletin	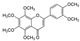	-	−1.98
Sinensetin	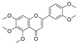	−2.06	−3.68
Tangeritin	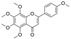	−2.84	−3.33

**Table 2 molecules-27-02467-t002:** *h*MAOs inhibitory activities of bergamottin and the reference MAO inhibitors rasagiline (*h*MAO-B) clorgyline for (*h*MAO-A).

Compound	2D Structure	IC_50_ (µM)	SI
*h*MAO-A	*h*MAO-B
Bergamottin	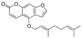	9.25 ± 0.91	0.291 ± 0.032	31.8
Clorgyline		0.00260 ± 0.00033	1.93 ± 0.16	0.00135
Rasagiline		3.72 ± 0.38	0.149 ± 0.023	24.9

## Data Availability

All data are available in this publication.

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
