# Peer review of "Molecular Modeling and Experimental Evaluation of Non-Chiral Components of Bergamot Essential Oil with Inhibitory Activity against Human Monoamine Oxidases"

_molecules, 2022, doi:10.3390/molecules27082467_

Round 1
Reviewer 1 Report
Ii is good study , it is involved new study about ((Molecular Modeling and Experimental Evaluation of Non-Chi- 2 ral Components of Bergamot Essential Oil with Inhibitory Ac- 3 tivity Against Human Monoamine Oxidases))..
but it needs minor corrections like :
1- there is no compared between current results and prevousily results in other studies to improve current results .
2-The conclusions are not covering all results in this study .
3- how do you chose this data (IC50 (µM) )) write method ...
4- in any university or scientific -Lab ,,, the study carried out ?
5- I accepted paper after corrections
Author Response
Point 1: there is no compared between current results and prevousily results in other studies to improve current results.
Response 1: The Bergamottin MAO inhibition is an absolute novelty, no literature data are available. Anyway in paragraph 2.2. Bergamottin inhibition data are compared to standard MAOs inhibitors such as Clorgyline and Rasagiline.
Point 2: The conclusions are not covering all results in this study
Response 2: According to the reviewer suggestion, the Conclusions section has been widely modified.
Point 3: how do you chose this data (IC50 (µM) )) write method ...
Response 3: The paragraph 4.2 report experimental details that should clarify how IC50 data were determined. Actually, IC50 values were obtained from dose-response curves and were expressed as mean standard deviation. IC50 values were computed from at least three independent experiments, each performed in triplicate.
Point 4: in any university or scientific -Lab ,,, the study carried out ?
Response 4: As reported by the Authors' affiliations, the study is a collaboration among two Universities (Catanzaro and Porto), one academic spinoff company (Net4Science) and one Research institute (CRISEA).
Reviewer 2 Report
In this manuscript, the authors reported an important topic about the plausible inhibitors for hMAO-A and hMAO-B. The 4 selected inhibitors are bergamottin, nobiletin, sinensetin and tangeritin which are nature products from Bergamot Essential Oil (BEO). This research is based on molecular docking and In vitro monoamine oxidase inhibition studies (IC50) value. Authors provided comprehensive introduction for the scientific background. However, I don’t think this manuscript can be published on Molecules in the current form. I think this study is far from finished. As you claimed in the introduction section, it just “providing a rational basis for further investigation”. Here, I still have some suggestions for your research. (1) Molecular docking can only provide rough qualitative results and initial binding structures. It’s not enough for providing quantitative discussions. Different scoring functions may provide different results. To include quantitative discussion, force field-based MD simulation is needed. FF-MD simulation is a widely used method. You can used FEP to calculate the binding free energy (J. Chem. Theory Comput. 2021, 17, 7, 4262–4273). (2) Please provide more binding structural insights. Like this in (Nature volume 469, pages175–180 (2011)), (In attached file) The pi-pi stacking conclusion lacks of evidence. By the way, energy decomposition analysis can provide deeper insight. These insights may be important for drug discovery. (3) The reasons why you research non-chiral compounds needs to be elaborated. Is “Complication in lead optimization”(page 2 line 90) a scientific reason or a technical reason? Non-chiral compounds from BEO may lack of studies. After all, it’s an interesting topic. Wish your further updates.

Author Response
First of all, we would like to thank the reviewer for his productive suggestions. We made our best to follow them and we guess the manuscript has improved its overall quality.
Point 1: Molecular docking can only provide rough qualitative results and initial binding structures. It’s not enough for providing quantitative discussions. Different scoring functions may provide different results. To include quantitative discussion, force field-based MD simulation is needed. FF-MD simulation is a widely used method. You can used FEP to calculate the binding free energy (J. Chem. Theory Comput. 2021, 17, 7, 4262–4273).
Response 1: We agree with the suggestion and Molecular Dynamics simulation have been performed and included in the manuscript.
Point 2: Please provide more binding structural insights. Like this in (Nature volume 469, pages175–180 (2011)), (FIGURE NOT DISPLAYED HERE), The pi-pi stacking conclusion lacks of evidence. By the way, energy decomposition analysis can provide deeper insight. These insights may be important for drug discovery.
Response 2: Figure 1 was completely re-drawn accordingly to the suggestion. It now reports the final structure from molecular dynamics simulations. Moreover, the ligand-targets interactions, computed on the molecular dynamics trajectories, are displayed in a schema.
Point 3: The reasons why you research non-chiral compounds needs to be elaborated. Is “Complication in lead optimization”(page 2 line 90) a scientific reason or a technical reason? Non-chiral compounds from BEO may lack of studies.
Response 3: To better explain our rationale, the following sentence was added to paragraph 2.1: “Already known MAOIs and chiral compounds were excluded from the selection. In fact, chiral molecules introduce uncertainty if they are not available as pure enantiomers: molecular modeling results will be related to the single enantiomer while experimental data will be derived from mixture.”
Reviewer 3 Report
In this study, the authors selected a novel selective and reversible inhibitor against human monoamine oxidases (MAO) using docking software. And then, the selective MAO-B inhibitory activity of the compound bergamottin was validated. They found that the compound is favorable to bind with MAO-B. Although the scientific question is highly relevant, the study lack of the originality. Moreover, some simulation methods are missing. The results are not shown. Therefore, it is not suitable for the Molecules journal.
Author Response
Point 1: In this study, the authors selected a novel selective and reversible inhibitor against human monoamine oxidases (MAO) using docking software. And then, the selective MAO-B inhibitory activity of the compound bergamottin was validated. They found that the compound is favorable to bind with MAO-B. Although the scientific question is highly relevant, the study lack of the originality.
Response 1: We have better clarified the goal of the present study has been the discovery of a novel activity related to the Bergamottin. The following sentences have been included in the Abstract and in Conclusions:
Abstract: "In agreement with the computational results, experimental studies confirmed both the never previously reported hMAO inhibition properties of Bergamottin and its preference for the isoform B."
Conclusions: "Finally, this study discovered a scientifically unknown activity of Bergamottin suggesting new tentative application for BEO"
Moreover, the Conclusion paragraph has been almost completely re-written for highlight the tentative role of Bergamot in neurodegenerative disorders.
Point 2: Moreover, some simulation methods are missing. The results are not shown. Therefore, it is not suitable for the Molecules journal.
Response 2: The molecular modeling part of the manuscript has been largely modified. Molecular Dynamics simulations have been carried out. More details, both related to new and previous calculations, have been included in the Materials and Methods section. All available data have been reported.
In light of all modifications we have done, we hope the reviewer could re-consider his last sentence.
Reviewer 4 Report
In the paper entitled “Molecular Modeling and Experimental Evaluation of Non-Chiral Components of Bergamot Essential Oil with Inhibitory Activity Against Human Monoamine Oxidases”, Catalano et al. evaluated in silico and in vitro the inhibitory activity of several non-chiral components of BEO on human monoamine oxidases. Please find below my comments, suggestions, and recommendations.
Major issue:
- The results are not discussed at all and are not interpreted in the perspective of previous studies.
Minor issue:
Introduction:
- Line 85: ”In silico” must be written in italic (see lines 128 and 184)
Materials and methods:
Molecular docking:
- Line 147: Please provide the CIDs (entries) of the compounds retrieved from PubChem
- Line 155: Please provide the size (in A3) of the search box
- Line 159: The authors should mention the ranking criteria (the theoretical binding energy)
In vitro monoamine oxidase inhibition studies
- Line 163: The company from which the authors obtained the cell line should be mentioned
- Line 170: The authors need to provide the source of bergamottin (it was purchased or chemically synthesized?) as well as for the MAO inhibitors used (clorgyline and rasagiline).
Results and discussions
- Table 1 and 2. Insert the 2D structures of the compounds
Author Response
We would like to thank the reviewer for his revision which helped us in improving the manuscript.
Point 1: The results are not discussed at all and are not interpreted in the perspective of previous studies.
Response 1: Other simulations have been carried out. In this light discussion and conclusion have been widely enriched.
Point 2: Line 85: ”In silico” must be written in italic (see lines 128 and 184)
Response 2: the text has been modified according to the suggestion
Point 3: Line 147: Please provide the CIDs (entries) of the compounds retrieved from PubChem
Response 3: CIDs have been included in paragraph 4.1
Point 4: Line 155: Please provide the size (in A3) of the search box
Response 4: the search box volume, equal to 64000 A3, has been reported in paragraph 4.1
Point 5: Line 159: The authors should mention the ranking criteria (the theoretical binding energy)
Response 5: Both paragraphs 2.1 and 4.1 have been modified. Now it should be more clear that we used the default Glide scoring function for ranking docked compounds.
Point 6: Line 163: The company from which the authors obtained the cell line should be mentioned
Response 6: at the end of paragraph 4.2 we clarify that MAO enzymes (obtained from cell lines) were purchased from Sigma Aldrich
Point 7: Line 170: The authors need to provide the source of bergamottin (it was purchased or chemically synthesized?) as well as for the MAO inhibitors used (clorgyline and rasagiline).
Response 7: all compounds have been purchased. At the end of paragraph 4.2 the name of all vendors has been included
Point 8: Table 1 and 2. Insert the 2D structures of the compounds
Response 8: probably some technical problem occurred. 2D structures were present in the uploaded manuscript but disappeared in the document for the reviewers. Anyway, all 2D structures have been in the revised manuscript.
Round 2
Reviewer 2 Report
The soundness of this research is improved after that reversion.